# Effectiveness of Fish Roe, Snow Fungus, and Yeast Supplementation for Cognitive Function: A Randomized, Double-Blind, Placebo-Controlled Clinical Trial

**DOI:** 10.3390/nu15194221

**Published:** 2023-09-29

**Authors:** Yung-Kai Lin, Yung-Hsiang Lin, Chi-Fu Chiang, Li Jingling

**Affiliations:** 1Institute of Food Safety and Risk Management, National Taiwan Ocean University, Keelung 20224, Taiwan; yklin@ntou.edu.tw; 2Department of Food Science, National Taiwan Ocean University, Keelung 202301, Taiwan; 3Graduate Institute of Biomedical Engineering, National Chung Hsing University, Taichung 40227, Taiwan; 4Research & Design Center, TCI Co., Ltd., Taipei 11494, Taiwan; vincent@tci-bio.com (Y.-H.L.); jimmy.chiang@tci-bio.com (C.-F.C.); 5Graduate Institute of Biomedical Sciences, China Medical University, Taichung 40402, Taiwan

**Keywords:** cognition, short-term memory, fish roe, tremella fuciformis, yeast

## Abstract

The brain is one of the most critical organs in the human body, regulating functions such as thinking, memory, learning, and perception. Studies have indicated that fish roe, snow fungus, and yeast may have the potential to modulate cognitive, memory, and emotional functions. However, more relevant clinical research in this area still needs to be conducted. This study explored the cognition-enhancing potential of a formula beverage including fish roe, snow fungus, and yeast. Sixty-four subjects were divided into a placebo group (*n* = 32) and a formula-drink group (*n* = 32), who consumed the product for 8 weeks. Cognitive tests were administered and analyzed at weeks 0, 4, and 8. After 4 and 8 weeks, there was a significant increase in the number of memory cards, and the response times among those who consumed the formula beverage were significantly faster than those in the placebo group. The subjects remembered the old items better and were more impressed with similar items based on the week effect. There was a significant increase in the cue effect of happy facial expressions after the subjects consumed the formula beverage for 8 weeks. In addition, there was a significant decrease in anxiety and fatigue, and improved quality of life. This formula beverage is a promising option that could be used to prevent further cognitive decline in adults with subjective cognitive complaints.

## 1. Introduction

In recent years, with the increasing aging population and mounting life stress, there has been a growing focus on the need to protect the brain and improve cognitive function. As one of the most vital organs in the human body, the brain plays a crucial role in regulating complex cognitive processes such as thinking, memory, learning, and perception. Extensive research has indicated that factors such as adequate nutrition, moderate exercise, sufficient sleep, and psychological well-being significantly contribute to brain functioning and cognitive abilities [1]. Neurons serve as the fundamental functional units of the brain, and they communicate with each other through chemical and electrical signals, forming an intricate neural network [2]. The connectivity and activity within this neural network are essential for cognitive function. The connections between neurons form neural networks and transmit information through synapses. The strength and plasticity of these synapses, referring to their ability to change and adapt, are crucial for memory formation and the learning process [3]. Neurotransmitters play a pivotal role in regulating the transmission of information between neurons. Common neurotransmitters, such as dopamine, acetylcholine, and glutamate, participate in the modulation of attention, emotions, and memory, among other cognitive functions [4]. As the pursuit of brain health continues, there is a growing interest in brain-health supplements. These supplements encompass products that are extensively advertised as providing nutritional support, promoting health, and improving specific functions, including brain health. These products claim to enhance memory, increase focus, improve cognitive function, and delay brain aging, among other benefits. However, due to the highly complex nature of brain functioning, which involves various neurotransmitters, neuronal connections, neural plasticity, and physiological mechanisms, it is challenging for single-component supplements to yield definitive effects on brain function. Furthermore, most brain-health supplements need more clinical trials and scientific-research support.

Fish roe, also known as fish eggs, is regarded as a delicacy in many cultures, and it is famous for its rich nutritional content. Fish roe is believed to contain a range of essential nutrients, including deoxyribonucleic acid (DNA)**,** omega-3 fatty acids, vitamins, minerals, and antioxidants, which are considered crucial for brain function and cognitive health [5]. Deoxyribonucleic acid (DNA) and its metabolic derivatives play crucial roles in numerous physiological functions. The addition of DNA to food is beneficial for immune function and cellular tissue growth, development, and repair. The brain is a highly complex organ that requires various nutrients to maintain its optimal functioning. Among these nutrients, omega-3 fatty acids, such as docosahexaenoic acid (DHA) and eicosapentaenoic acid (EPA), are essential for brain health [6]. These fatty acids are believed to support the structure and function of brain cells, facilitate communication between neurons, and reduce inflammation in the brain. They are associated with cognitive-function, memory, and mood-regulation improvements. In addition to omega-3 fatty acids, fish roe is a rich source of vitamins and minerals essential for brain health [7]. These include B vitamins, such as B12 and folate, which are crucial for producing and maintaining neurotransmitters involved in cognitive function. Minerals like iron, zinc, and selenium participate in various processes within the brain, including oxygen transport, antioxidant defense, and the synthesis of neurotransmitters [8]. Snow fungus, called Tremella fuciformis, is a natural edible and medicinal mushroom commercially available in Taiwan and Asian countries. It is renowned for its unique nutritional and medicinal value and is believed to have particular benefits for brain health and cognitive function. Snow fungus contains various nutrients, including proteins, dietary fiber, vitamins, and minerals. It also contains some specific bioactive compounds, such as polysaccharides and peptides, which are thought to play a positive role in brain health [9]. Research has shown that the polysaccharides in snow fungus have antioxidant and anti-inflammatory properties, which can reduce oxidative stress and inflammation, protecting brain cells from damage [10]. These properties may contribute to the prevention of neurodegenerative diseases, such as Alzheimer’s disease and Parkinson’s disease. Additionally, the polysaccharides in snow fungus are believed to promote the growth and regeneration of nerve cells, helping to maintain normal neuronal function and cognitive abilities [11]. Snow fungus also contains certain plant compounds, such as polyphenols and triterpenoids, which possess antioxidant and anti-inflammatory properties, providing additional protection to brain cells [12]. Based on the above studies, it can be stated that fish roe and snow fungus protect the brain. Yeast powder, also known as yeast-fungus powder or brewer’s yeast powder, is a common food additive and nutritional supplement. It is made from yeast that undergoes fermentation and drying and possesses rich nutritional value and various bioactive components. Research indicates that the B vitamins, particularly vitamin B6, vitamin B12, and folic acid, found in yeast powder are essential for normal brain function [13]. These vitamins participate in the synthesis and regulation of neurotransmitters, aiding in the proper functioning and communication of nerve cells. They also play essential roles in energy metabolism and DNA synthesis, contributing to the growth and repair of nerve cells [8]. However, few clinical studies have been conducted on fish roe, snow fungus, and yeast powder.

This study investigates the potential of a formula beverage containing fish roe, snow fungus, and yeast as the main ingredients to enhance cognitive function. Figure 1 illustrates the summary design and findings of the current study. The 64 subjects were divided into a placebo group (*n* = 32) and a formula-beverage group (*n* = 32), and they were administered the respective treatments for 8 weeks. Cognitive tests were conducted and analyzed at weeks 0, 4, and 8.

## 2. Materials and Methods

### 2.1. Clinical Trial Design

This study is a double-blind, placebo-controlled clinical trial. The Institutional Review Board of China Medical University Hospital (CMUH, Taichung, Taiwan) approved the study (CMUH111-REC3-127), and written informed consent was obtained from the subjects. The ClinicalTrials.gov identifier for this study is NCT05988593. For this study, we recruited 64 healthy adults. The inclusion criteria were healthy adult males or females aged 20 to 65. The exclusion criteria were: (1) breastfeeding, pregnant or planning to become pregnant during the trial (self-reported), (2) heart, liver, kidney, endocrine, or other major organic diseases (self-reported), (3) long-term medication use (self-reported), (4) neuropsychiatric disorders or brain surgery, (5) inability to see 12-point font on a computer clearly after vision correction. Subjects were randomly assigned to either the formula-beverage or placebo group. In total, 64 subjects were recruited, with 32 placed in the Formula beverage and 32 in the placebo group. The manufacturer kept the unblinding forms of the numbers, obtained after the experiment’s completion, for data analysis.

### 2.2. Test Sample

Formula beverage (MelaGene+™, Melaleuca, Shanghai, China) contained 11.3% (3400 mg; the deoxyribonucleic acid (DNA) content of roes is 31.06 mg/mL) salmon roe, 0.3% (90 mg) yeast, 0.18% (56 mg) snow fungus, citric acid, and water. Placebo beverage’s main ingredients were citric acid and water. Subjects consumed a bottle of drink (30 g) daily and underwent cognitive testing at weeks 0, 4, and 8.

### 2.3. Match-to-Sample Test

The test assessed motor control, attention, memory, and executive function, linked to the activity level of the lateral intraparietal area (LIP) in monkey studies [14]. Subjects needed to match pairs of pictures as quickly as possible within one minute. The maximum number of pairs completed was their score for the match-to-sample memory task. The test lasted approximately 5 min.

### 2.4. The Trail-Making Test

This is a well-known neuropsychological test that assesses motor control, attention switching, planning, execution ability, and the ability to monitor the process of achieving a plan. It is effective in detecting patients with frontal-lobe syndrome in clinical settings and is widely used in clinical problems such as depression [15], attention-deficit hyperactivity disorder [16], stroke [17], and Alzheimer’s disease [18]. The task is intuitive, with the first version requiring subjects to connect dots in numerical order and the second version requiring subjects to switch between two sequences, such as connecting 1-A-2-B in order. Subjects must complete the connection as quickly as possible, with completion time being their score for the task. This test lasts approximately 10 min.

### 2.5. The Processing-Speed Test

This test measures the time it takes for a participant to make a corresponding hand response after seeing a visual stimulus. Processing speed typically decreases with age, with peak performance occurring in early adulthood. It may also slow down when individuals are tired or as a result of aging [19,20]. In our test, three rows of buttons are presented, with stimuli randomly appearing in one row. Subjects have 30 s to press the corresponding button as quickly as possible, with the number of successful button presses being their score. This test lasts approximately 5 min.

### 2.6. Mnemonic Similarity Test (MST)

We used the MST to assess hippocampal function. Participants indicated whether objects were “old”, “similar”, or “new” by pressing a button on a computer [21]. Responses to lure items were of particular interest, with a “similar” response indicating successful pattern separation and an “old” response indicating a bias toward pattern completion [22]. The task stimulates hippocampal function and takes approximately 10 min to complete.

### 2.7. State-Trait Anxiety Inventory (STAI)

The STAI is a subjective anxiety-assessment, including anxiety traits (anxiety trait Y2) and anxiety states (anxiety state Y1), over two assessments, with a total of forty questions, rated on a four-point scale [23]. The lowest scores for anxiety state Y1 are 20 points and the highest score is 80 points, with clinical significance reached above 40 points. Anxiety trait Y2 is also rated in the range of 20–80 points; usually, scores of more than 40 points indicate the need for professional assistance. It takes about 5 min to fill out the questionnaire.

### 2.8. Facial-Expression-Cuing Task (FECT)

The facial-expression-cuing task and attention test assesses participants’ ability to evaluate emotions in faces, using angry, happy, and sad faces as cue images. Previous research has shown that individuals with poor emotional control are more easily attracted by emotional stimuli [24]. This test was therefore used to evaluate the effect of the drink on emotional control and took approximately 10 min to complete.

### 2.9. Assessment of Perceived Stress

Fatigue scale (PSS-14) is a subjective fatigue-assessment scale, with fourteen questions with five options each, which can assess the subjective fatigue levels of subjects [25]. This scale is included as an evaluation tool for fatigue. This scale lasts about 3 min.

### 2.10. Quality-of-Life-Assessment Scale

The WHOQOL-BREF is a simplified quality-of-life assessment scale developed by the World Health Organization. The original version of the questionnaire was provided by the WHO and was approved for utilization and translation into Chinese. The WHOQOL-BREF has 28 questions covering overall physiological, psychological, social, and environmental assessments. It is widely used and has cross-national validity. The scale lasts about 5 min.

### 2.11. Statistical Analysis

The comparison of measurement results for cognitive parameters within groups and between groups was analyzed by two-way mixed ANOVA followed by SPSS 12.0, and *p* < 0.05 was considered statistically significant.

## 3. Results

### 3.1. Formula Beverages Increased Cognitive Function

Table 1 shows the numbers of subjects and their ages in both groups, with no significant differences in baseline at week 0. Therefore, the background information of the subjects in both groups was homogeneous. Table 2 displays the results of three cognitive-function tests: match-to-sample, trail-making, and processing-speed. The match to sample refers to the number of cards that can be remembered within a time limit, serving as an indicator of short-term memory. After 4 and 8 weeks, there was a significant increase in the number of memory cards (*p* = 0.049) among the participants who consumed the formula beverage compared to the placebo group. Furthermore, after 8 weeks, there was a significant increase in the number of memory cards that could be remembered by the participants consuming the formula beverage compared to the placebo group (Figure 2A). The connectivity task is divided into two sections: the trail-making test Section 1 (TMT1) and the trail-making test Section 2 (TMT2). The TMT1 assesses abilities such as motor control, sequencing, and short-term memory. The TMT2 builds upon TMT1 by adding abilities such as cognitive switching, planning, execution, and monitoring. After 4 and 8, the response times in both TMT1 and TMT2 were significantly faster among those who consumed the formula beverage compared to the placebo group (*p* < 0.001) (Figure 2B,C). Processing-speed testing assesses the ability to apply responses within a certain time frame flexibly. Both groups increased the number of keys and did not differ (Figure 2D).

### 3.2. Formula Beverage Improved Memory

The subjects then underwent the mnemonic similarity test, which consisted of two phases. The first stage involved a category assessment, while the second stage assessed the subjects’ recollections of previous categories. Each subject’s ability to distinguish similar items was used to indicate memory. Table 3 indicates the accuracy of the item-categorization results from the first stage, while the accuracy of the memory recall (divided into accuracy for new items, old items, and similar items) was the result of the second stage. The results show that the accuracy for new items was high, the accuracy for old items improved with practice, and the accuracy for similar items was the lowest. To further analyze whether the ability to remember was altered by drinking the beverages, we used two indicators: recollection of old items and similar impression scores. Table 3 shows that after taking the formula beverage for 8 weeks, the subjects remembered the old items better (*p* = 0.015) and were more impressed with similar items (*p* = 0.024) based on the week effect (Figure 3).

### 3.3. Formula Beverage Improved Emotions

Next, facial expressions depicting three emotions, sadness, happiness, and anger, were used to analyze the categories and intensity of emotions. Table 4 shows the results of the emotional assessment of the subjects and the attentional effect of using emotional faces as cues through facial-expression-cueing tasks. The emotional assessments were recorded in the form of questionnaires, and the approach was to use emotional faces as clues to attract the subjects’ attention, in order to test the degree to which their attention was attracted. After taking the formula beverage for 8 weeks, there was a significant increase in the perceptions of facial emotions (*p* = 0.032). In contrast, in the placebo group, these perceptions decreased (Figure 4A), indicating that consuming formula beverages enhances the ability to perceive emotions and leads to a stronger perception of emotions displayed on others’ faces. Regarding the response times, a stronger cue effect indicated that attention is more easily drawn to that cue. However, in terms of accuracy, the opposite was true. A higher value indicated that less attention was paid to the cue. After the formula-beverage subjects consumed the beverage for 4 weeks, there was a significant increase in the cue effect of happy facial expressions (*p* = 0.039), while the placebo group showed a decrease (Figure 4B). This indicates that consuming the formula beverages enhanced the propensity to pay attention to faces displaying happiness, thereby increasing the cue effect. The Inverted Efficiency Score (IES) considers the possibility of trading response time for accuracy. It corrects for situations in which accuracy is sacrificed for speed and calculates the cue effect accordingly. A higher IES indicates that attention is more easily drawn to a cue. Table 4 shows that after consuming the formula beverage for 8 weeks, there was a significant increase in the cue effect of happy facial expressions (*p* = 0.046). This indicates that consuming the formula beverage increased the attention towards positive emotional facial expressions.

### 3.4. Formula Beverage Improved Anxiety and Quality of Life

Table 5 indicates the results of the subjects’ anxiety, stress, and quality-of-life measures. After the subjects consumed the formula beverage for 8 weeks, there was a significant decrease in anxiety, in terms of both anxiety states (*p* = 0.045) and anxiety traits (*p* = 0.047) which was based on the week effect (Figure 5A,B). After the subjects consumed the formula beverage for 8 weeks, fatigue was slightly improved (Figure 5C), and improved quality of life was observed (*p* = 0.044) (Figure 5D).

## 4. Discussion

Based on the current research findings, some dietary supplements may have certain improvement effects on cognitive and memory functions. However, further scientific research and clinical trials are needed to ascertain the reliability of these effects and elucidate their specific mechanisms. In this clinical study, a formula beverage containing a high dose of fish roe, snow fungus, and yeast was used. It was found that the consumption of this beverage led to improvements in cognitive functions, memory, and emotional well-being.

Fish roe may potentially affect cognition, memory, mood, and anxiety symptoms. Fish roe is a nutrient-rich food that contains various substances, such as omega-3 fatty acids, protein, vitamin D, and vitamin B12 [26]. These nutrients play essential roles in normal brain function. Fish roe contains omega-3 long-chain polyunsaturated fatty acids (LC-PUFAs), including eicosapentaenoic acid (EPA) and docosahexaenoic acid (DHA), which are crucial for brain formation and function. The dietary intake of EPA and DHA is often reflected by the omega-3 index in the blood [27]. Epidemiological studies have shown associations between lower omega-3 levels and adverse outcomes, including reduced brain volume, cognitive impairment, and mental disorders [28]. Omega-3 fatty acids are essential dietary fats that play significant roles in the function of the nervous system, vision, the immune system, cardiovascular health, the skin, and connective tissues. A necessary omega-3 PUFA, DHA, must be obtained through the diet [29]. In the adult brain, PUFAs account for approximately 20% of the fatty-acid composition in the gray matter. If the diet lacks diversity, dietary sources of DHA, such as salmon and mackerel, may not provide optimal levels of DHA absorption. As an omega-3 fatty acid, DHA has led to dose-dependent improvements in learning and memory abilities in rats. The mechanism behind this improvement involves the upregulation of NR1, CREB, and c-fos-mRNA expression in the rat hippocampus [30,31]. In mice, DHA has also been found to enhance learning and memory function, as demonstrated through the Morris water-maze test [32]. Additionally, DHA administration reduced neuronal damage in mouse brains and increased unsaturated-fatty-acid levels while decreasing saturated-fatty-acid levels. Notably, DHA inhibited the deposition of amyloid-beta (Aβ) plaques and tau-protein neurofibrillary tangles, which are associated with neurodegenerative disorders [32]. The mechanism of action behind these effects is attributable to the upregulation of β-secretase (BACE)2 expression, which competes with BACE1 in cleaving amyloid precursor protein (APP), resulting in the reduced production of extracellular Aβ fragments [33].

The fish roe in this trial product contained higher DNA contents (31.06 mg/mL). Salmon-roe DNA has been reported to improve brain function in mice [34,35]. A study of gene expression in the hippocampus of the brain showed increased gene expressions of markers for neurons, microglia, astrocytes, and oligodendrocytes, indicating that DNA ingestion may be involved in promoting differentiation into the various cells of the brain. This suggests that the increase in nucleic acids containing cytosine bases in the hippocampus is involved in the improvement of memory and learning ability [34]. Salmon-roe DNA has also been reported to prevent liver inflammation [36]. Another study showed a link between chronic inflammation and cognitive decline and neurodegenerative diseases, such as Alzheimer’s disease [37]. Salmon-roe DNA may be an anti-inflammatory ingredient derived from a natural food component that inhibits brain inflammation, which is increased by stress and aging.

The relationship between plant-based foods and cognitive function is also a focus of attention. Plant-based foods, such as chia seeds, flaxseeds, and walnuts, also contain alpha-linolenic acid (ALA), which is a type of omega-3 fatty acid. Studies have shown that ALA may contribute to cognitive health [7]. A further study indicated that higher intake of ALA is associated with better cognitive function in older adults [38]. Fruits and vegetables rich in polyphenolic compounds are beneficial for improving cognitive function and preventing cognitive decline [39]. The DASH (Dietary Approaches to Stop Hypertension) and MIND (Mediterranean–DASH Diet Intervention for Neurodegenerative Delay) diets both emphasize the consumption of plant-based foods. These diets are associated with a lower risk of cognitive decline and neurodegenerative diseases [40]. Tremella fuciformis extracts contain polysaccharides, including glucuronic acid, xylose, and fucose [41]. Studies have shown that the intake of polysaccharides can improve cognition and memory. Although the detailed mechanism behind this is still unclear, relevant studies have pointed out that Tremella fuciformis extracts can scavenge reactive oxygen species, regulate immunity, and promote neurodevelopment, including the stimulation neurite growth, the promotion of long-term memory in the hippocampus, increasing the release of acetylcholine, and reducing the scopolamine-induced impairment of learning and memory in rats [42]. A treatment with Tremella fuciformis (100 or 400 mg/kg) orally for 14 consecutive days significantly reversed scopolamine-induced deficits in learning and memory and alleviated the decrease in cholinergic immunoreactivity induced by scopolamine in the medial septum and hippocampus [42]. The results of another study suggest that the promotion of neuritogenesis in neuronal culture cells using Tremella fuciformis water extracts is associated with the ability to improve rats’ performances on spatial learning and memory tasks [42]. Moreover, impairments in spatial learning and memory may be attributed to a decrease in the activation of the septohippocampal cholinergic system, and Tremella fuciformis partially ameliorated learning and memory deficits by increasing central cholinergic activity [42].

Yeast and fermentation are necessary processes involved in the making of food. Yeast is a single-celled fungus that plays a central role in the fermentation process. During fermentation, yeast converts sugars and carbohydrates in food into alcohol and carbon dioxide. This process is widely used in the production of food and beverages, contributing to the creation of a wide variety of products. Fermented plant-based foods, such as fermented vegetables and fruits, play a crucial role in human nutrition by providing vitamins, minerals, trace elements, and other essential nutrients. The effects of supplementation with Lactobacillus helveticus IDCC3801-mediated fermented milk (LHFM) on cognitive function in healthy older adults were studied in subjects who received supplements with different concentrations of LHFM for 12 weeks [43]. The results of the cognitive tests showed that taking LHFM for three months improved cognitive function in healthy older adults, as confirmed. Significant changes were found in biomarkers (BDNF and WBV) and self-rating scales (PSS and GDS-SF). The results of another study indicated that the consumption of L. helveticus CM4-mediated fermented milk effectively improved the cognitive function of middle-aged healthy Japanese adults [44]. Furthermore, the effects of supplementation with probiotic yeast-fermented milk (Bifidobacterium bifidum, Lactobacillus casei, Lactobacillus fermentum, and Lactobacillus acidophilus; dose: 2 × 109 CFU per gram of each probiotic per day) (for 12 weeks) on Alzheimer’s disease were studied. A 12-week intervention showed that the probiotic milk positively affected cognitive function in patients with Alzheimer’s disease. Similarly, according to our results, formula beverages containing fish roe, Tremella fuciformis, and yeast can improve cognitive, memory, and emotional functions.

## 5. Conclusions

This study shows that drinking formula beverages containing high doses of fish roe, Tremella fuciformis, and yeast can effectively improve short-term memory and executive performance, lead to cognitive transformation, reduce anxiety, and increase sensitivity to the emotions of others. The safety and tolerance profiles of the beverage are favorable. Thus, this formula beverage is a promising option that could be used to prevent further cognitive decline in adults with subjective cognitive complaints.

## Figures and Tables

**Figure 1 nutrients-15-04221-f001:**
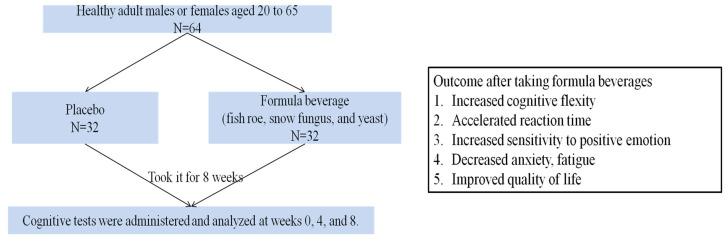
A graphic summary of this study.

**Figure 2 nutrients-15-04221-f002:**
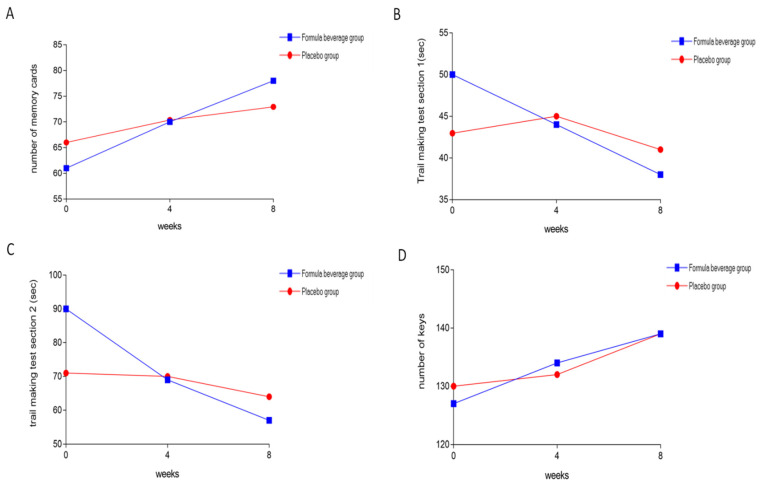
Formula beverages increase cognitive function. The 64 subjects were divided into a placebo group (*n* = 32) and a formula-beverage group (*n* = 32), and they were administered the respective treatments for 8 weeks. (**A**) The match-to-sample test was conducted and analyzed at weeks 0, 4, and 8. (**B**,**C**) The trail-making test Section 1 (TMT1) and the trail-making test Section 2 (TMT2) were conducted and analyzed at weeks 0, 4, and 8. (**D**) The processing-speed test was conducted and analyzed at weeks 0, 4, and 8.

**Figure 3 nutrients-15-04221-f003:**
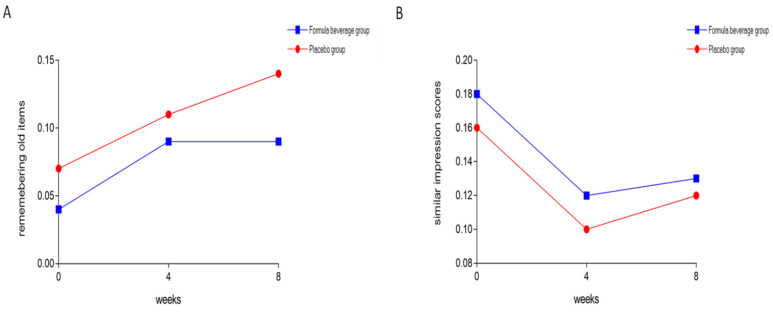
Formula beverage improved memory. The 64 subjects were divided into a placebo group (*n* = 32) and a formula-beverage group (*n* = 32), and they were administered the respective treatments for 8 weeks. Subjects underwent a mnemonic similarity test, which consisted of two phases. (**A**) The first stage involved a category assessment, (**B**) while the second stage assessed the subjects’ recollections of previous categories.

**Figure 4 nutrients-15-04221-f004:**
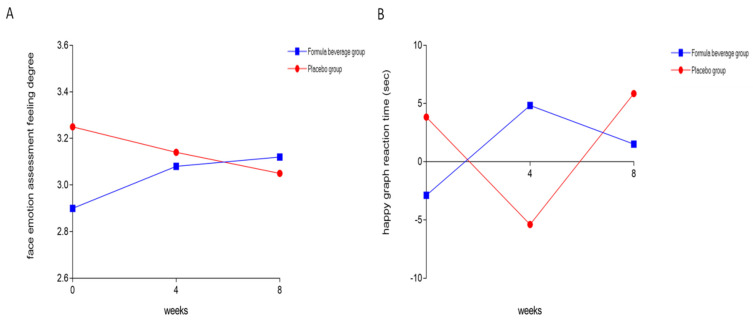
Formula beverage improved emotions. The 64 subjects were divided into a placebo group (*n* = 32) and a formula-beverage group (*n* = 32), and they were administered the respective treatments for 8 weeks. Subjects underwent facial-expression-cuing tasks, which included (**A**) the facial emotion assessment and (**B**) the emotional attention test.

**Figure 5 nutrients-15-04221-f005:**
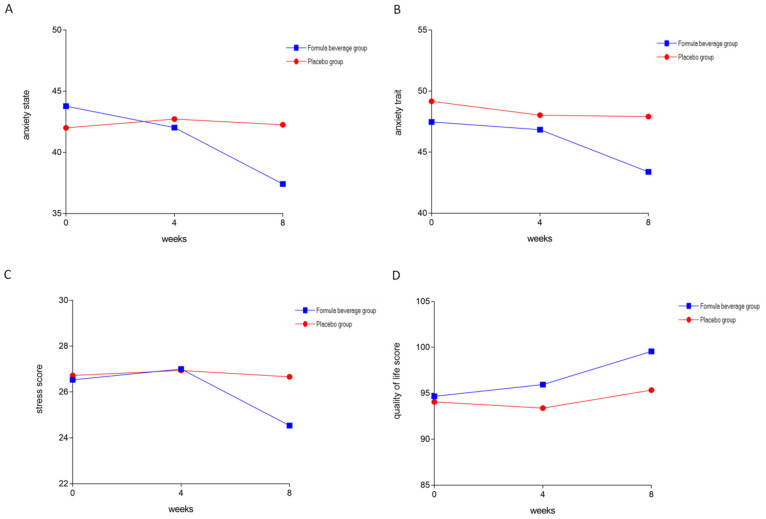
Formula beverage improved anxiety and quality of life. The 64 subjects were divided into a placebo group (*n* = 32) and a formula-beverage group (*n* = 32), and they were administered the respective treatments for 8 weeks. Subjects were assessed with (**A**,**B**) anxiety scale, (**C**) stress scale, and (**D**) quality-of-life scale.

**Table 1 nutrients-15-04221-t001:** Basic information of subjects.

	Formula-Beverage Group	Placebo Group
Number	32	32
Male	9	11
Female	23	21
Average age	28.93	28.18
Maximum age	59.22	56.47
Minimum age	20.18	20.44

**Table 2 nutrients-15-04221-t002:** Results of cognitive-function tests (match-to-sample, trail making, and processing speed).

Index		Number of Memory Cards	SD	Trail-Making Test Section 1	SD	Trail-Making Test Section 2	SD	Number of Keys	SD
Formula beverage	week 0	61.69	±23.08	50.35	±17.39	90.54	±30.78	127.41	±39.63
	week 4	70.44	±23.85	44.4	±15.51	69.68	±21.97	134.88	±42.32
	week 8	78.19	±26.96	38.89	±11.70	57.91	±16.09	139.78	±43.44
Placebo beverage	week 0	66.31	±19.63	42.97	±9.95	71.54	±20.71	130.09	±40.70
	week 4	70.38	±25.24	45.71	±11.32	70.74	±18.71	132.72	±42.35
	week 8	72.94	±25.31	41.04	±10.15	64.64	±15.80	139.25	±40.12
Week 0, 4, 8	between-group effect	–		–		–		–	
	week effect	<0.001		<0.001		<0.001		<0.001	
	interaction	0.049		<0.001		<0.001		–	
Week 0, 4	between-group effect	–		–		–		–	
	week effect	–		<0.001		<0.001		<0.001	
	interaction	–		–		<0.001		–	
Week 4, 8	between-group effect	–		–		–		–	
	week effect	0.01		<0.001		<0.001		<0.001	
	interaction	–		–		–		–	

SD: standard deviation.

**Table 3 nutrients-15-04221-t003:** Results of the mnemonic similarity task.

Index		Accuracy of Item Categorization	SD	Accuracy of Memory Recall (New Items)	SD	Accuracy of Memory Recall (Old Items)	SD	Accuracy of Memory Recall (Similar Items)	SD	Remembering Old Items	SD	Similar Impression Scores	SD
Formula beverage	week 0	0.93	±0.08	0.74	±0.16	0.65	±0.17	0.47	±0.19	0.04	±0.22	0.18	±0.20
	week 4	0.94	±0.06	0.72	±0.15	0.69	±0.18	0.49	±0.22	0.09	±0.25	0.12	±0.18
	week 8	0.94	±0.05	0.72	±0.18	0.7	±0.21	0.49	±0.18	0.09	±0.29	0.13	±0.21
Placebo beverage	week 0	0.94	±0.04	0.72	±0.13	0.67	±0.15	0.47	±0.13	0.07	±0.17	0.16	±0.16
	week 4	0.94	±0.04	0.69	±0.11	0.7	±0.17	0.48	±0.16	0.11	±0.21	0.10	±0.14
	week 8	0.94	±0.05	0.72	±0.14	0.73	±0.18	0.47	±0.15	0.14	±0.24	0.12	±0.15
Week 0, 4, 8	between-group effect	–		–		–		–		–		–	
	week effect	–		–		<0.001		–		0.015		0.024	
	interaction	–		–		–		–		–		–	
Week 0, 4	between-group effect	–		–		–		–		–		–	
	week effect	–		–		0.005		–		0.036		0.003	
	interaction	–		–		–		–		–		–	
Week 4, 8	between-group effect	–		–		–		–		–		–	
	week effect	–		–		–		–		–		–	
	interaction	–		–		–		–		–		–	

SD: standard deviation.

**Table 4 nutrients-15-04221-t004:** Results of facial emotion assessment and emotional attention.

		Emotional Assessment				Cue Effect																	
Index		Accuracy of Facial Emotion Assessment	SD	Facial Emotion Assessment Feeling Degree	SD	Angry Graph Reaction Time	SD	Happy Graph Reaction Time	SD	Sad Graph Reaction Time	SD	Accuracy of Anger Graph	SD	Accuracy of Happy Graph	SD	Accuracy of Sad Graph	SD	Angry Ies	SD	Happy Ies	SD	Sad Ies	SD
Formula beverage	week 0	0.75	±0.11	2.9	±0.46	−2.45	±0.02	−2.88	±0.03	−1.4	±0.03	0	±0.05	6.02	±0.05	−15.63	±0.08	−4.97	±0.04	−6.35	±0.04	10.19	±0.05
	week 4	0.79	±0.13	3.08	±0.47	−5.54	±0.02	4.83	±0.02	−1.18	±0.02	−1.95	±0.05	24.41	±0.06	6.84	±0.04	−4.8	±0.04	−10.1	±0.05	−5.54	±0.03
	week 8	0.78	±0.11	3.12	±0.67	0.68	±0.02	1.51	±0.02	−9.71	±0.02	5.86	±0.05	−5.86	±0.06	−1.95	±0.05	−2.11	±0.04	5.03	±0.0.05	−8.76	±0.04
Placebo beverage	week 0	0.8	±0.05	3.25	±0.48	4.59	±0.03	3.83	±0.02	−1.04	±0.02	4.88	±0.06	6.84	±0.05	−5.86	±0.05	1.76	±0.05	1.25	±0.03	1.75	±0.04
	week 4	0.8	±0.06	3.14	±0.57	−1.92	±0.03	−5.39	±0.02	−5.65	±0.02	−3.91	±0.05	−2.93	±0.05	1.95	±0.05	0.29	±0.03	−4.9	±0.03	−6.57	±0.04
	week 8	0.76	±0.15	3.05	±0.72	−1.28	±0.03	5.85	±0.02	−9.5	±.0.04	−7.81	±0.04	−2.93	±0.06	−6.84	±0.05	4.15	±0.05	7.68	±0.03	−7.63	±0.06
Week 0, 4, 8	between-group effect	–		–		–		–		–		–		–		–		–		–		–	
	week effect	–		–		–		–		–		–		–		–		–		–		–	
	interaction	–		0.032		–		–		–		–		–		–		–		–		–	
Week 0, 4	between-group effect	–		–		–		–		–		–		–		–		–		–		–	
	week effect	–		–		–		–		–		–		–		–		–		–		–	
	interaction	–		–		–		0.039		–		–		–		–		–		–		–	
Week 4, 8	between-group effect	–		–		–		–		–		–		–		–		–		–		–	
	week effect	–		–		–		–		–		–		–		–		–		0.046		–	
	interaction	–		–		–		–		–		–		–		–		–		–		–	

SD: standard deviation.

**Table 5 nutrients-15-04221-t005:** Results of the anxiety scale, stress scale, and quality-of-life scale.

Index		Anxiety State	SD	Anxiety Trait	SD	Stress Score	SD	Quality of Life Score	SD
Formula beverage	week 0	43.78	±10.74	47.47	±11.93	26.53	±9.19	94.66	±18.14
	week 4	42.03	±10.11	46.84	±11.96	27.03	±8.01	95.94	±15.44
	week 8	37.41	±9.16	43.38	±11.19	24.53	±8.35	99.56	±15.83
Placebo beverage	week 0	42.06	±11.35	49.16	±10.72	26.72	±6.98	94.06	±13.69
	week 4	42.72	±10.38	48.03	±10.48	26.94	±6.61	93.38	±15.34
	week 8	42.25	±9.97	47.91	±9.79	26.66	±7.99	95.34	±12.86
Week 0, 4, 8	between-group effect	–		–		–		–	
	week effect	0.045		0.047		–		0.044	
	interaction	0.034		–		–		–	
Week 0, 4	between-group effect	–		–		–		–	
	week effect	–		–		–		–	
	interaction	–		–		–		–	
Week 4, 8	between-group effect	–		–		–		–	
	week effect	–		–		–		<0.001	
	interaction	–		–		–		–	

SD: standard deviation.

## Data Availability

The datasets analyzed or generated during this study are available from the corresponding author on reasonable request.

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
