# Peer review of "Effectiveness of Fish Roe, Snow Fungus, and Yeast Supplementation for Cognitive Function: A Randomized, Double-Blind, Placebo-Controlled Clinical Trial"

_nutrients, 2023, doi:10.3390/nu15194221_

Round 1

Reviewer 1 Report

This manuscript was objected to investigate fish roe, fungus and yeast supplementation on cognitive effect. The clinical results should be presented by appropriate statistical method. However, in this manuscript, although the authors used paired t-test and one-way ANOVA, the presented statistics was interactions. This results seems to be confusing. In addition, all the standard deviations should be presented in the whole results.

Before clinical trial, index or active compounds and the chemical profiles should be analyzed. Main active compound should be discussed.

English is good, however minor English editing should be proceeded.

Author Response

Thank you very much for your precious suggestions. We have modified our manuscript in a point-by-point manner.

Reviewer 1:

This manuscript was objected to investigate fish roe, fungus and yeast supplementation on cognitive effect. The clinical results should be presented by appropriate statistical method. However, in this manuscript, although the authors used paired t-test and one-way ANOVA, the presented statistics was interactions. This results seems to be confusing. In addition, all the standard deviations should be presented in the whole results.

ANS: Thank you for pointing out the mistake. We ran a mixed two-way ANOVA rather than a one-way ANOVA or a paired t-test. The two factors for ANOVA were group (formula drink group vs. placebo group, a between factor) and weeks (0, 4, or 8 weeks after taking the beverage, a within factor). In this case, a significant interaction means that group differences can be observed in certain weeks. In the previous version, we inadvertently misapplied the analytical method, which has been rectified in the current version. Please see page 5, line 191, as follows:

The comparison of measurement results for cognitive parameters within groups and between groups was analyzed by two-way mixed ANOVA followed by SPSS, as p < 0.05 was considered statistical significance.

Also, as suggested, the standard deviations were added in all the Tables in the whole results.

Before the clinical trial, the index or active compounds and the chemical profiles should be analyzed. The main active compound should be discussed.

ANS: This formula beverage was analyzed before we sent it to clinical trials. The supposed active compounds are listed on page 3, Lines 128-132. The test report is attached on the next page. The main compound found is DNA. Meanwhile, the beverage was produced with fish roe, yeast, and snow fungi. We are unsure which compound is the main active compound to the cognitive function improvement we observed, so we included discussions on possible contributions from all the compounds in the Discussion.

Reviewer 2 Report

Thank you for involving me in the review of this trial that investigated the efficacy of fish egg, snow fungus, and yeast supplementation on cognitive function.

the topic is very interesting and noteworthy.

The formatting of the text needs to be revised because the font size is not uniform in the abstract.

Ethical issues: has the study been approved by the ethics committee? Report the approval code and provide a copy of the proceedings to the editorial office.

How was the age range chosen? why 20-65? justify the choice.

Line 325: Implement discussion of plant-based foods and cognitive function. Refer to studies on the topic

Providing a graphic abstract would support the reader.

Author Response

Reviewer 2:

Thank you for involving me in the review of this trial that investigated the efficacy of fish egg, snow fungus, and yeast supplementation on cognitive function. The topic is very interesting and noteworthy.

ANS: Thank you for your recognition of our research.

The formatting of the text needs to be revised because the font size is not uniform in the abstract.

ANS: Thank you for indicating this point. The formatting was transferred automatically from the website, and we did not notice there were font size variation problems. We carefully reviewed the text and unified font size and format in the revision.

Ethical issues: has the study been approved by the ethics committee? Report the approval code and provide a copy of the proceedings to the editorial office.

ANS: Yes, this study was approved by the ethics committee in CMUH (CMUH111-REC3-127) and was written on page 3, lines 115-116. A copy of the approval is attached here.

How was the age range chosen? why 20-65? justify the choice.

ANS: The present study aims to recruit healthy adults as participants. Given that the legal definition of adulthood in our country begins at the age of 20, we have established a minimum age criterion of twenty years old. Furthermore, considering the well-documented decline in cognitive function with increasing age, it is commonly accepted that individuals aged 65 and above are considered to be in the elderly stage. Consequently, we have set an upper age limit of 65 years old for adult participants in this study.

By the way, the legal definition of adulthood was changed to 18 years old in December 2022 in Taiwan. While we designed participants from September 2021 and started recruiting participants in September 2022, we decided to stick to the old criterion.

Line 325: Implement discussion of plant-based foods and cognitive function. Refer to studies on the topic

ANS: Thank you for your kind suggestions. We have put two paragraphs in the Discussion on plant-based food and cognitive functions (pages 11-12). Most studies showed possible contributions to learning and memory, and some to preventing neurodegenerative diseases.

Providing a graphic abstract would support the reader.

ANS: Thank you for your constructive suggestion. We have plotted a graphic abstract and inserted it as Figure 1 in the text as follows.

Round 2

Reviewer 1 Report

The revised manuscript was well addressed for reviewer's comments. 

Reviewer 2 Report

Manuscript well improved